# Narrative Voice Matters! Improving Smoking Prevention with Testimonial Messages through Identification and Cognitive Processes

**DOI:** 10.3390/ijerph17197281

**Published:** 2020-10-05

**Authors:** Juan-José Igartua, Laura Rodríguez-Contreras

**Affiliations:** Department of Sociology and Communication, Faculty of Social Sciences, Campus Unamuno (Edificio FES), 37007 Salamanca, Spain; laurarodriguezcontreras@usal.es

**Keywords:** narrative persuasion, smoking prevention, testimonial messages, narrative voice, identification with the protagonist, cognitive processes

## Abstract

Narrative messages are increasingly being used in the field of tobacco prevention. Our study is based on narrative persuasion and aims to analyze the psychological mechanisms that explain why the narrative voice is relevant to promote persuasive impact. An online experiment with a 2 (narrative voice) × 2 (message) factorial design was carried out. Participants (525 adult smokers) were randomly assigned to two experimental conditions (first-person versus third-person narrative message). To increase the external validity of the study, two different messages were used within each condition. After reading the narrative message the mediating and dependent variables were evaluated. Participants who read the narrative in the first person experienced greater identification. Moreover, mediational analysis showed that both counterarguing and cognitive elaboration played a significant role in the relationship between narrative voice, identification, and persuasive impact. This study confirm that narrative voice is not only an anecdotal formal choice but that it indirectly affects variables related to tobacco prevention, due to the fact that first-person messages activate a mechanism of affective connection with the message (increasing the identification with the protagonist) that decreases resistance to prevention (the counterarguing process) while simultaneously stimulating reflection or cognitive elaboration.

## 1. Introduction

Tobacco is one of the main public health problems facing humanity. The consumption of this substance has been linked to multiple health problems (such as respiratory and cardiovascular diseases and various forms of cancer), being responsible for more than 8 million deaths a year [1,2]. In Spain, 18.8% of women and 25.6% of men smoke daily, constituting the second most widespread psychoactive substance in the population and being responsible for 50,000 deaths per year (13% of all deaths) [3]. However, smoking is also the largest preventable cause of premature death and illness [4]. Therefore, improving the effectiveness of smoking prevention campaigns or campaigns aimed at helping smokers to quit is an important health communication goal.

The design of campaigns for the prevention of smoking requires innovative approaches that consider both the characteristics of the messages and the psychological processes they trigger. In this sense, narrative messages are increasingly being used in the field of tobacco prevention [5,6]. (An example of narrative intervention for tobacco prevention is the *Tips from Former Smokers* campaign, developed by the Centers for Disease Control and Prevention (CDC) in the USA (https://www.cdc.gov/tobacco/campaign/tips/index.html). In this context, our research focusses on testimonial messages or personal narratives, delivered by an adult smoker who tells their story regarding successful tobacco cessation. This type of narrative message focusses on specific individual cases, adopts an experiential style and does not include general, abstract or statistical information [7]. The ultimate goal of these types of messages is to cause a persuasive impact. In the present work, we use the expression persuasive impact to refer to four outcome measures: intention to quit smoking, perceived effectiveness of the message, expectations of self-efficacy (confidence in one’s ability to abstain from smoking) and expectations of the efficacy of the preventive response (response efficacy or outcome expectations). These measures constitute relevant variables in the theoretical models on health communication and behavioral change (e.g., theory of planned behavior, theory of reasoned action, health belief model, social cognitive theory and extended parallel process model; for a review, see [8]). The mentioned theoretical models “are frequently applied to the crafting of persuasive health messages and campaigns” [9]. Furthermore, these measures have been used as dependent variables in several previous studies on smoking prevention using narrative messages [5,6,10].

Meta-analyses have shown that narrative interventions produce significant effects, but it is also observed that not all narratives are effective [11,12]. Therefore, it is important to determine which ingredients of such narratives are most effective from a persuasive point of view. In this sense, in the present work, we focus on one characteristic of narrative messages that can condition their persuasive efficacy: narrative voice [13,14].

The narrative voice is a property of the text, being defined as the perspective adopted when telling a story, among which messages written in the first, second or third person can be differentiated [15,16,17,18]. In first-person narratives, the narrator is inside the story and directly expresses their views on a subject or their experience. In this way, the audience has access to the protagonist’s internal life, which facilitates engagement with the protagonist. On the other hand, messages written in the third person include a narrator who relates the experience of the protagonist from an external or spectator’s perspective. Therefore, third-person messages establish a kind of *mental firewall* that separates the reader from the psychological life of the protagonist of the story. Finally, second-person narratives identify the reader of the narrative as the protagonist, but they are rarely used in health campaigns [19].

Despite being a very relevant formal resource for the design of this type of message [19,20], analysis of the effect of narrative voice has received insufficient attention in research on narrative persuasion for smoking control. For example, in a review of 153 experimental studies on health-related narrative persuasion, only 4 manipulated this attribute of the message, and only 1 of them focused on smoking prevention [19]. Our work therefore aims to contribute to knowledge on the impact of narrative voice in smoking prevention, as this line of research has been developed very little to date.

Narrative voice constitutes a formal characteristic of the message that determines the relationship between the audience and the protagonist of the story. In this context, a second objective of this work is to analyze the psychological mechanisms that explain why narrative voice constitutes a relevant element to promote persuasive impact. Based on the main theoretical models on narrative persuasion [21,22,23], several relevant mechanisms are established in the current study. 

Previous research has found that people better imagine the thought processes of protagonists (e.g., “Did you *see* what the main character was thinking and seeing?”) when narrative messages are written in the first compared with the third person [24]. Therefore, it has been proposed that first-person narratives make it easier for the audience to identify with the protagonist [16,20,25]. In this context, identification with the protagonist (or the capacity to feel and adopt the point of view of the protagonist of the narrative; [14,26,27] is proposed as a *primary mediator*. It is assumed that messages in the first person enhance the aforementioned process, to a greater extent than messages in the third person, by helping the individual to adopt the protagonist’s perspective and better imagine their thought processes, thus leading to our first hypothesis:

**Hypothesis** **H1:**
*Compared with a third-person message, a first-person narrative message will induce greater identification with the protagonist.*


The main theoretical models of narrative persuasion consider that identification fosters persuasive impact, since this process inhibits resistance to the persuasive proposal of the message, facilitating attitudinal impact [22,23]. However, the empirical evidence in this regard is inconclusive [28]. Furthermore, since identification constitutes a process of *temporal involvement* with the message [26], it is also possible that it will increase the cognitive elaboration during the reception of the message [29].

In this context, it is assumed that identification can facilitate persuasive impact through three cognitive processes that would act as *secondary mediating* mechanisms. Consistent with the Entertainment Overcoming Resistance Model (EORM) [22], identification can be expected to reduce counterarguing (the production of critical cognitive responses that refute the content of the message; [30]) and reactance (negative reactions to the message when perceiving that freedom of choice or opinion is being threatened [31]). Furthermore, it may be expected that identification will be associated with an increase in cognitive elaboration (reflecting on the topic of the message during its processing) [29,32]. However, to date, the role of counterarguing, reactance and cognitive elaboration has not been analyzed simultaneously within a single mediational model, so our study constitutes an original contribution to this field. Therefore, our work tries to contrast the role of these three cognitive mechanisms to explain the indirect effect of narrative voice on measures related to the prevention of smoking, leading to the following mediational hypothesis (Figure 1):

**Hypothesis** **H2:**
*The indirect effect of narrative voice on the perceived effectiveness of the message, on the intention to quit smoking, on the self-efficacy expectations and on the response efficacy expectations will be serially mediated by the identification with the protagonist and by the cognitive processes of counterarguing (H2a), reactance (H2b) and cognitive elaboration (H2c) (Figure 1).*


## 2. Materials and Methods

### 2.1. Design and Participants Subsection

An online experiment with a 2 (narrative voice) × 2 (message) between-subjects factorial design was carried out. The participants were randomly assigned to two experimental conditions such that half read a narrative message written in the first person whereas the other half a message in the third person. Furthermore, to increase the external validity of the study, two different messages were used within each condition [33,34].

The online experiment was carried out using QUALTRICS to access an initial sample of 568 adult smokers. Of these, 43 were removed for failing the manipulation test (correctly remembering which type of message they had read: first or third person). The remaining sample consisted of 525 participants (50.9% women). Participant age ranged from 18 to 55 years old (*M* = 35.27, *SD* = 10.97) (see Table 1).

The applied questionnaire comprised three blocks: pre-test measures, reading of the narrative message (experimental manipulation) and post-test measures. Sociodemographic information was collected in the pre-test measures, and three screener questions were included to select the participants, such that only those who declared that they were current smokers, indicated having smoked more than 100 cigarettes during their life and had smoked 5 or more cigarettes every day during the previous week, were included. These eligibility criteria were used with reference to previous studies [6,35,36]. Moreover, the degree of tobacco dependence of the participants was also evaluated using the Fagerström test [37], revealing a moderate dependence on this substance (*M* = 4.65, *SD* = 2.23, *Md* = 5.00, on a scale with a theoretical range from 0 to 10). At the end of this block of questions, the participants were randomized to the experimental conditions, after which the post-test measures on the mediating processes (identification with the protagonist, counterarguing, reactance and cognitive elaboration) and the dependent variables (perceived effectiveness of the message, intention to quit smoking, self-efficacy expectations and response efficacy expectations) were presented.

QUALTRICS allows the implementation of a series of quality controls. The questionnaire was designed in such a way that it could only be completed in a single session. In addition, only those participants who took 6–45 min to complete the questionnaire (*M* = 10.95 min, *SD* = 4.99), took between 60 and 420 s to read the narrative (*M* = 112.88, *SD* = 53.58) and correctly answered an attention check question included in the questionnaire were counted as valid cases.

### 2.2. Independent Variable and Stimulus Materials

Taking as examples the testimonies of former smokers on forums and web pages as well as narratives used in previous studies [35], a written narrative delivered by a 45-year-old woman who indicated that she had quit smoking more than a year ago was constructed (narrative messages are available in the Appendix A file online, Appendix A). In her story, the protagonist alluded to topics such as the age at which she started smoking, why she considered quitting smoking (after having being diagnosed with periodontitis) and how she managed to quit tobacco, commenting in the final part of the narrative that she no longer wanted to start smoking again, that she had noted how the negative consequences associated with tobacco use had disappeared and that she was experiencing a number of benefits from quitting smoking.

In order to manipulate the narrative voice, the elements that mark the grammatical person in the written narratives were modified, such as the choice of personal pronouns [15,20]. In addition, in the first-person narrative, the protagonist introduced herself with her own voice and with her name at the beginning (“My name is Teresa, and I am 45 years old. I have been a smoker for 20 years but have not smoked for over a year.”) In contrast, in the third-person narrative, an external observer introduced the protagonist and told the story (“Teresa is 45 years old. She has been a smoker for 20 years but has not smoked for more than a year.”)

The two versions of the narrative message were identical in all other aspects of their content. Moreover, within each experimental condition (first versus third person), two different messages were used, differing from each other only regarding the information provided on the number of attempts the protagonist had made before quitting smoking. Thus, the messages emphasized that the protagonist had quit smoking on her first attempt (e.g., “She had not tried to quit before; this was her first time”) or fourth attempt (e.g., “She had already tried to quit smoking before, three times”). This subtle modification to the text allowed us to include more than one narrative message per experimental condition and thereby increase the external validity of the study, considering that people are often exposed to testimonies from former smokers with different profiles.

The four narratives used in the experiment had a similar length (between 425 and 428 words), which is the most common length used in this type of work [38]. A pilot study in which 105 people participated (67.6% women, *M* = 42.05 years, *SD* = 12.39) was carried out. The results showed that the designed narratives were perceived as clear and easy to understand, credible, interesting and realistic, with no statistically significant differences being observed between the four versions.

### 2.3. Measures

*Identification with the protagonist.* Identification was assessed using an 11-item scale [27] that measures the degree of identification with a specific character (e.g., “I felt as if I were Teresa”; from 1 = not at all to 5 = very much; α = 0.93, *M* = 3.60, *SD* = 0.84).

*Counterarguing.* A scale consisting of three items created from the counterarguing scale [28] was used (e.g., “While reading the story, I thought that I did not agree with some of the things said by Teresa”; from 1 = strongly disagree to 7 = strongly agree; α = 0.73, *M* = 2.72, *SD* = 1.26).

*Reactance.* This was evaluated using the perceived threat to freedom scale [39] consisting of four items (e.g., “The message tried to manipulate me”; from 1 = strongly disagree to 7 = strongly agree; α = 0.83, *M* = 2.64, *SD* = 1.37).

*Cognitive elaboration.* An adapted version of the cognitive elaboration scale [29] was used, consisting of three items (“While reading the narrative, I reflected intensely on the topic of tobacco use and its consequences”; from 1 = strongly disagree to 7 = strongly agree; *α* = 0.85, *M* = 5.33, *SD* = 1.26).

*Perceived effectiveness of the message.* This was evaluated using a scale constructed from previous work [6,10,40] and composed of four items (e.g., “the message was convincing”; from 1 = strongly disagree to 7 = strongly agree; α = 0.86, *M* = 5.27, *SD* = 1.14).

*Intention to quit smoking.* This was evaluated using a scale created from previous studies [5,6,35] and composed of three items (e.g., “it is very likely that I will quit smoking in the next 3 months”; from 1 = strongly disagree to 7 = strongly agree; α = 0.84, *M* = 4.72, *SD* = 1.46).

*Self-efficacy.* To measure expectations of self-efficacy, that is, the confidence of the participants in abstaining from smoking, a scale constructed from previous studies [41,42] and consisting of six items was used (e.g., “I think I have the capacity to stop smoking when I want to”; from 1 = strongly disagree to 7 = strongly agree; α = 0.87, *M* = 4.78, *SD* = 1.25).

*Response efficacy.* The expectations of efficacy of the preventive response (quitting smoking) were measured using a scale [25] comprising five items (e.g., “a life without tobacco reduces the risk of health problems”; from 1 = strongly disagree to 7 = strongly agree; α = 0.85, *M* = 5.91, *SD* = 0.97) (see Table 2).

### 2.4. Statistical Analysis

Data analyses were conducted using SPSS version 25, statistical software (IBM Company, Armonk, NY, USA). Descriptive analysis (means and standard deviations) were calculated to examine sample demographics (see Table 1). Reliability (Cronbach’s alpha) was calculated for all the measures (see Table 2). The correlations between the mediating and dependent variables were analyzed by using the Pearson correlation coefficient. One-way analysis of variance (ANOVA) and chi-squared test were used to test the success of randomization. Factorial ANOVA was performed to determine the impact of narrative voice on identification with the protagonist (H1), including the type of message as a second independent variable. Effect size in ANOVA test was calculated using partial eta-squared (partial *η*^2^); for nonsignificant results (*p*-values higher than 0.05), effect size was substituted by observed power (or post hoc power), as recommended by many statisticians (e.g., [43]; but also see [44]). To test hypothesis 2, the PROCESS macro (version 3.5) for SPSS developed by Hayes was used [45]. This macro makes it possible to test different mediational models based on the bootstrapping technique. According to the bootstrapping method, an indirect effect is statistically significant if the confidence interval established (*CI* at 95%) does not include the value 0. If the value 0 is included in the *CI*, the indirect effect is equal to 0, that is, there is no association between the variables considered.

## 3. Results

### 3.1. Preliminary Analysis

Randomization was successful: the conditions did not differ significantly on gender (χ^2^(3, *n* = 525) = 0.25, *p* = 0.969), age (*F* (3, 521) = 0.32, *p* = 0.809) or the degree of tobacco dependence (*F*(3, 521) = 0.67, *p* = 0.571).

The correlations between the mediating variables and the dependent variables were also analyzed. This analysis confirmed that the mediating processes showed convergent correlations with the proposed hypotheses (for example, between identification and counterarguing). In addition, it was also verified that the mediating processes showed statistically significant relationships with the dependent variables. These results justify the proposed mediational model (see Table 3).

### 3.2. Effect of Narrative Voice on Identification with the Protagonist (H1)

Analysis of variance (ANOVA) revealed that narrative voice significantly influenced the identification with the protagonist (*F*_narrative voice_(1, 521) *=* 6.59, *p*
_=_ 0.011, partial *η*^2^ = 0.013), while no statistically significant effects were observed for the type of message (*F*_message_(1, 521) = 2.16, *p* = 0.142, observer power = 0.31), nor interaction effects between narrative voice and message type (*F*_narrative voice × message_(1, 521) = 0.07, *p* = 0.781, observed power = 0.05). The results showed that people who read the narrative in the first person experienced greater identification with the protagonist (*M* = 3.69, *SD* = 0.81) than those who were exposed to the narrative in the third person (*M* = 3.51, *SD* = 0.85), thus confirming H1.

### 3.3. Testing a Serial-Parallel, Mediation Model (H2)

To test the second hypothesis, the PROCESS macro for SPSS (model 81, with 10,000 bootstrapping samples to generate 95% confidence intervals by the percentile method) was used [45]. The independent variable (narrative voice) was coded as a dummy variable (first-person message = 1, third-person message = 0), and the message-type variable was included in the analysis as a covariate. This procedure allowed the evaluation of the specific indirect effect of the experimental condition on the dependent variables through identification (as primary mediator) and cognitive processes (as secondary mediators).

It was observed that the narrative voice in the first person increased identification (*B* = 0.18, *SE* = 0.07, *p* = 0.010), and this in turn was associated with less counterarguing (*B* = −0.44, *SE* = 0.07, *p* = 0.000) and lower reactance (*B* = −0.34, *SE* = 0.07, *p* = 0.000), as well as greater cognitive elaboration (*B* = 1.08, *SE* = 0.04, *p* = 0.000). Lesser counterarguing was associated with higher perceived effectiveness of the message (*B* = -0.08, *SE* = 0.02, *p* = 0.000) and higher response efficacy expectations (*B* = −0.13, *SE* = 0.03, *p* = 0.000). Reactance was not statistically significantly associated with any of the dependent variables. In contrast, cognitive elaboration showed positive and statistically significant associations with the perceived effectiveness of the message (*B* = 0.35, *SE* = 0.03, *p* = 0.000), the intention to quit smoking (*B* = 0.48, *SE* = 0.05, *p* = 0.000), self-efficacy expectations (*B* = 0.15, *SE* = 0.06, *p* = 0.015) and response efficacy (*B* = 0.21, *SE* = 0.04, *p* = 0.000) (Figure 2).

Three statistically significant specific indirect effects of narrative voice were observed through identification on the perceived effectiveness of the message (*Effect* = 0.1154, *SE* = 0.0474, 95% *CI* (0.0271, 0.2119)), the intention to quit smoking (*Effect* = 0.0796, *SE*= 0.0376, 95% *CI* (0.0153, 0.1617)) and the response efficacy expectations (*Effect* = 0.0317, *SE* = 0.0187, 95% *CI* (0.0023, 0.0740)) (Table 3). In addition, two specific indirect effects were also observed, through the serial mediation of identification and counterarguing, on the perceived effectiveness of the message (*Effect* = 0.0074, *SE* = 0.0042, 95% *CI* (0.0009, 0.0171)) and the response efficacy expectations (*Effect* = 0.0114, *SE* = 0.0061, 95% *CI* (0.0020, 0.0255)), thus providing partial support to H2a. However, no significant specific indirect effects were observed through the serial mediation of identification and reactance on any of the dependent variables, which implies a rejection of H2b. Finally, four specific indirect effects were observed, through the serial mediation of identification and cognitive elaboration, on the perceived effectiveness of the message (*Effect* = 0.0725, *SE*= 0.0300, 95% *CI* (0.0167, 0.1348)), the intention to quit smoking (*Effect* = 0.0982, *SE*= 0.0414, 95% *CI* (0.0230, 0.1849)), self-efficacy expectations (*Effect* = 0.0312, *SE*= 0.0203, 95% *CI* ( 0.0004, 0.0776)) and response efficacy (*Effect* = 0.0439, *SE*= 0.0225, 95% *CI* (0.0080, 0.0954)), thus H2c receives empirical support (see Table 4).

## 4. Discussion

The results of the present experiment shed light on the mechanisms of narrative persuasion and highlighted the relevance of narrative voice for increasing the effectiveness of tobacco prevention interventions. In particular, the present work makes a significant contribution in this field and clarifies two important questions.

First, it is verified that it is possible to increase the identification with the protagonist by manipulating a *formal* mechanism in the message: the first-person narrative voice. Research in the field of narrative persuasion to analyze the effect of narrative voice has not received sufficient attention, despite its being a very relevant formal device for the design of narrative messages on tobacco prevention [20]. Thus, in the review presented by de Graaf et al. [19], from a total of 153 experimental studies on narrative persuasion related to health, only 4 that manipulated this attribute of the message were identified, and only 1 of them focused on smoking prevention. Furthermore, previous studies on the effect of narrative voice have not provided significant evidence in favor of messages written in the first person [15,17,20,25] or found that the effects were conditional (that is, other additional “ingredients” were needed in the message for the effect to occur [13]). For all these reasons, our work makes a significant contribution to the study of the effect of narrative voice in the context of smoking prevention.

Secondly, a serial mediation process is demonstrated, providing new knowledge about the relationship between identification and the cognitive processes of counterarguing and cognitive elaboration. On the one hand, through reduced counterarguing, the participants who identified more with the protagonist showed a more favorable reaction to the message and manifested a greater perceived efficacy of the preventive response (considering that quitting smoking would improve their personal health in the short and long term). This result is convergent with the postulates of the EORM model [22] and with the Extended Elaboration Likelihood Model [23] as well as some previous empirical studies [28,46,47]. In this way, our study reinforces the idea that the experience of *fusion* with the protagonist of the message becomes a process that is incompatible with a state of negative assessment (that would hinder prevention). However, it should be considered that reactance did not act as a significant mediating mechanism, despite its negative correlation with identification (*r* = −0.21, *p* = 0.000) and with cognitive elaboration (*r* = - 0.15, *p* = 0.000), and a positive, strong and statistically significant correlation with counterarguing (*r* = 0.43, *p* = 0.000).

However, our research reveals that identification, which is conceived as a state of *temporal involvement* [26], is also associated with greater cognitive elaboration and that this reflective process increases the persuasive efficacy of the message. It should be noted that the role of cognitive elaboration in narrative persuasion processes has been less widely investigated to date compared with the role of message resistance processes (such as counterarguing or reactance) [48]. Therefore, our work makes a significant contribution to understanding the role of cognitive processes in smoking prevention using testimonial messages or personal narratives. Thus, it is verified that identification can stimulate *dual cognitive processing*, so that both processes (the reduction of *counterarguing* and the increase in elaboration) can act in tandem to achieve a persuasive impact through testimonial narrative messages.

The most important limitation of this work is that the proposed mediators were measured rather than being experimentally manipulated, which prevents conclusions with total certainty regarding the proposed causal sequence (identification → counterarguing → outcomes; identification → reactance → outcomes; identification → cognitive elaboration → outcomes). This problem is present in studies that contrast serial mediational models in this field [5,36]. Although temporal precedence is an important element to establish a causal inference, it is also necessary to propose a theoretical argument for the relationship between the mediating mechanisms, a condition that our work fulfils by relying on theoretical models of narrative persuasion. Indeed, future research should use other methodological approaches to deal with such causal inference problems [49].

## 5. Conclusions

The results of the present study lead us to raise two issues with important theoretical implications. First, identification with a positive role model who is the protagonist of a testimonial message designed to prevent smoking makes it difficult to produce negative responses to the message. Indeed, it is perhaps difficult to counterargue against the *biography* of a person who reports having successfully quit tobacco. Secondly, the experience narrated by this positive role model in the message can *inspire* and stimulate deep cognitive processing in people, so that they question and *adjust* their opinions about tobacco.

Beyond the theoretical implications, our work shows that certain characteristics of testimonial messages aimed at smoking prevention play a primary role in increasing their persuasive impact. In this work, it is shown that narrative voice is not only an *anecdotal formal choice*, but that it indirectly affects variables related to tobacco prevention, due to the fact that first-person messages activate a mechanism of affective connection with the message (identification with the protagonist) that reduces resistance to prevention while simultaneously stimulating reflection.

The results obtained in our research suggest various applications in the field of prevention and treatment of smoking. We consider that a narrative intervention such as that proposed (through narrative messages written in the first person, with testimonies of former smokers who relate their successful experience) could constitute a primary prevention tool, since any attempt to reduce tobacco consumption indirectly seeks to avoid the development of associated health problems. This type of testimonial messages (relating the experience of someone who has overcome tobacco addiction) would thus serve both to prevent smoking (stopping people from starting to use tobacco) and to help active smokers to quit tobacco, thus avoiding the damage that it may cause to them. In this sense, testimonial messages of former smokers could be used in prevention campaigns and be disseminated through social networks (for example, Instagram or Facebook), or as support materials in broader campaigns disseminated on web pages.

## Figures and Tables

**Figure 1 ijerph-17-07281-f001:**
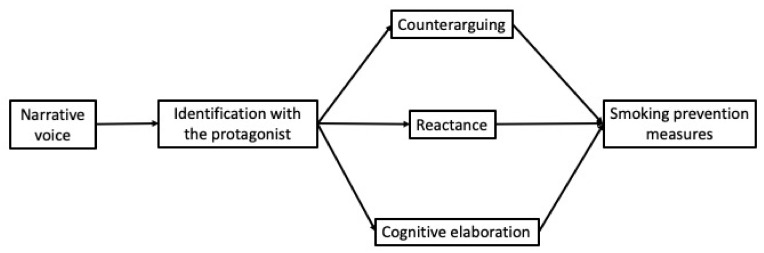
Hypothesized serial–parallel mediation model (H2).

**Figure 2 ijerph-17-07281-f002:**
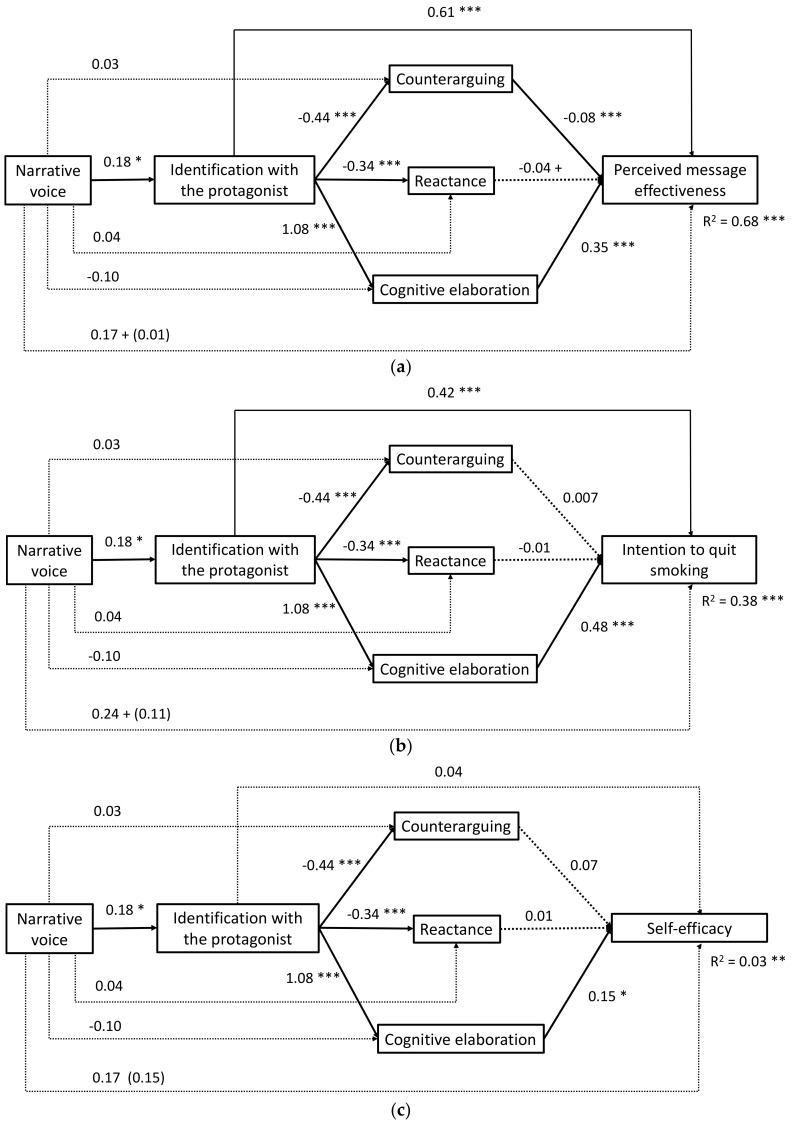
Results of the mediation analysis (H2). The figures show the non-standardized regression coefficients, *B*. The coefficients of the direct effects appear in parentheses. The dashed line represents nonsignificant coefficients. + *p* < 0.10, * *p* < 0.05, ** *p* < 0.01, *** *p* < 0.001. (**a**) Dependent variable: *perceived message effectiveness*. (**b**) Dependent variable: *intention to quit*. (**c**) Dependent variable: *self-efficacy*. (**d**) Dependent variable: *response efficacy*.

**Table 1 ijerph-17-07281-t001:** Characteristics of the study participants (*n* = 525).

	Mean (SD) or Percentage	Range
Age	*M = 35.27*	18–55
	*SD = 10.97*	
Sex	*Male: 258 (49.1%)*	
	*Female: 267 (50.9%)*	
Fagerström test	*M = 4.65*	0–10
	*SD = 2.23*	

**Table 2 ijerph-17-07281-t002:** Key measures.

Measure	Response Options	Reliability (Cronbach’s alpha)
Identification with the protagonist	1 (not at all)–5 (very much)	0.93
I felt emotionally involved with Teresa’s feelings I felt as if I were Teresa I imagined how I would act if I were Teresa I was concerned about what was happening to Teresa I understood how Teresa acts, thinks and feels I experienced Teresa’s emotional reactions myself I tried to imagine Teresa’s feelings, thoughts and reactions I had the impression of living Teresa’s story myself I understood Teresa’s feelings or emotions I tried to see things from Teresa’s point of view I identified with Teresa		
Counterarguing	1 (strongly disagree)–7 (strongly agree)	0.73
While reading the story, I thought that I did not agree with some of the things said by Teresa While reading the story, I thought that the information given by Teresa was inaccurate, misleading or exaggerated While reading the story, I tried to determine whether there were errors in Teresa’s conclusions on some topics		
Reactance	1 (strongly disagree)–7 (strongly agree)	0.83
The message threatened my freedom of choice The message tried to make a decision for me The message tried to manipulate me The message tried to pressure me		
Cognitive elaboration	1 (strongly disagree)–7 (strongly agree)	0.85
While reading the narrative, I reflected intensely on the issue of tobacco use and its consequences As I progressed through the narrative, I tried to draw conclusions to adjust my views on tobacco Reading the message has made me think deeply about what a life without tobacco would be like		
Perceived effectiveness of the message	1 (strongly disagree)–7 (strongly agree)	0.86
The message was believable The message was convincing This message has been very important to me Reading this message helped me feel more confident about dealing with tobacco Reading the message, I have been concerned about my smoking habit		
Intention to quit smoking	1 (strongly disagree)–7 (strongly agree)	0.84
I’m thinking I’m going to make an effort to quit smoking It is very likely that I will quit smoking in the next 3 months I will definitely quit smoking in the future		
Self-efficacy	1 (strongly disagree)–7 (strongly agree)	0.87
I think I have the ability to quit smoking when I want to I’m sure I can quit smoking I know what I should do to quit smoking If I quit smoking and someone offered me a cigarette, I would know how to resist and would not smoke If I quit smoking and attended a party with friends or family, I would know how to act in order not to smoke If I have already decided not to smoke again, I am sure I would not take a cigarette, even if I felt sad or anxious		
Response efficacy	1 (strongly disagree)–7 (strongly agree)	0.85
I am convinced that, if I stop smoking, my health will improve shortly thereafter I am sure that, if I stop smoking, my body will soon recover from the harmful effects of tobacco I am convinced that, if I stop smoking, it will decrease my risk of serious illnesses in the future Even if you have been smoking for many years, it is possible to become healthy again if you stop smoking in time A life without tobacco reduces the risk of health problems		

**Table 3 ijerph-17-07281-t003:** Descriptive analysis and correlations between mediating and dependent variables.

	1	2	3	4	5	6	7	8
1 Identification	-	-	-	-	-	-	-	-
2 Counterarguing	−0.29 ***	-	-	-	-	-	-	-
3 Reactance	−0.21 ***	0.43 ***	-	-	-	-	-	-
4 Cognitive elaboration	0.71 ***	−0.25 ***	−0.15 ***	-	-	-	-	-
5 Perceived message effectiveness	0.77 ***	−0.35 ***	−0.24 ***	0.74 ***	-	-	-	-
6 Intention to quit smoking	0.54 ***	−0.17 ***	−0.12 **	0.59 ***	0.60 ***	-	-	-
7 Self-efficacy	0.11 **	0.03	0.01	0.15 ***	0.17 ***	0.36 ***	-	-
8 Response efficacy	0.41 ***	−0.22 ***	−0.22 ***	0.43 ***	0.47 ***	0.58 ***	0.21 ***	-
Mean	3.60	2.72	2.64	5.33	5.27	4.72	4.78	5.91
Standard deviation	0.84	1.26	1.37	1.26	1.14	1.46	1.25	0.97

Note. *n = 525.* For all the variables, a higher score indicates a greater intensity of the considered process, from 1 for “low” to 7 to “high” (except for the identification scale, which has a theoretical range from 1 for “low” to 5 for “high”). ** *p* < 0.01, *** *p* < 0.001.

**Table 4 ijerph-17-07281-t004:** Specific indirect effects of narrative voice on perceived message effectiveness, intention to quit smoking, self-efficacy and response efficacy through identification and cognitive processes (H2). Mediation models with PROCESS.

**(a) Dependent variable: *perceived message effectiveness***			
**Specific indirect effects (mediators)**	**Effect**	**Boot SE**	**Boot 95% CI**
Narrative voice → Identification → Perceived message effectiveness	**0.1154**	**0.0474**	**[0.0271, 0.2119]**
Narrative voice → Counterarguing → Perceived message effectiveness	−0.0028	0.0101	[−0.0241, 0.0169]
Narrative voice → Reactance → Perceived message effectiveness	−0.0018	0.0058	[−0.0147, 0.0092]
Narrative voice → Cognitive elaboration → Perceived message effectiveness	−0.0384	0.0275	[−0.0925, 0.0168]
Narrative voice → Identification → Counterarguing → Perceived message effectiveness (H2a)	**0.0074**	**0.0042**	**[0.0009, 0.0171]**
Narrative voice → Identification → Reactance → Perceived message effectiveness (H2b)	0.0028	0.0023	[−0.0002, 0.0085]
Narrative voice → Identification → Cognitive elaboration → Perceived message effectiveness (H2c)	**0.0725**	**0.0300**	**[0.0167, 0.1348]**
**(b) Dependent variable: *intention to quit smoking***			
**Specific indirect effects (mediators)**	**Effect**	**Boot SE**	**Boot 95% CI**
Narrative voice → Identification → Intention to quit smoking	**0.0796**	**0.0376**	**[0.0153, 0.1617]**
Narrative voice → Counterarguing → Intention to quit smoking	0.0002	0.0053	[−0.0102, 0.0126]
Narrative voice → Reactance → Intention to quit smoking	−0.0007	0.0060	[−0.0143, 0.0120]
Narrative voice → Cognitive elaboration → Intention to quit smoking	−0.0520	0.0390	[−0.1334, 0.0209]
Narrative voice → Identification → Counterarguing → Intention to quit smoking (H2a)	−0.0006	0.0044	[−0.0095, 0.0087]
Narrative voice → Identification → Reactance → Intention to quit smoking (H2b)	0.0010	0.0034	[−0.0056, 0.0085]
Narrative voice → Identification → Cognitive elaboration → Intention to quit smoking (H2c)	**0.0082**	**0.0414**	**[0.0230, 0.1849]**
**(c) Dependent variable: *self-efficacy***			
**Specific indirect effects (mediators)**	**Effect**	**Boot SE**	**Boot 95% CI**
Narrative voice → Identification → Self-efficacy	0.0079	0.0226	[−0.0374, 0.0557]
Narrative voice → Counterarguing → Self-efficacy	0.0024	0.0097	[−0.0161, 0.0245]
Narrative voice → Reactance → Self-efficacy	0.0005	0.0063	[−0.0124, 0.0147]
Narrative voice → Cognitive elaboration → Self-efficacy	−0.0165	0.0156	[−0.0536, 0.0062]
Narrative voice → Identification → Counterarguing → Self-efficacy (H2a)	−0.0063	0.0056	[−0.0200, 0.0019]
Narrative voice → Identification → Reactance → Self-efficacy (H2b)	−0.0008	0.0035	[−0.0081, 0.0065]
Narrative voice → Identification → Cognitive elaboration → Self-efficacy (H2c)	**0.0032**	**0.0203**	**[0.0004, 0.0776]**
**(d) Dependent variable: *response efficacy***			
**Specific indirect effects (mediators)**	**Effect**	**Boot SE**	**Boot 95% CI**
Narrative voice → Identification → Response efficacy	**0.0317**	**0.0187**	**[0.0023, 0.0740]**
Narrative voice → Counterarguing → Response efficacy	−0.0043	0.0148	[−0.0330, 0.0268]
Narrative voice → Reactance → Response efficacy	−0.0024	0.0077	[−0.0205, 0.0117]
Narrative voice → Cognitive elaboration → Response efficacy	−0.0232	0.0175	[−0.0599, 0.0098]
Narrative voice → Identification → Counterarguing → Response efficacy (H2a)	**0.0114**	**0.0061**	**[0.0020, 0.0025]**
Narrative voice → Identification → Reactance → Response efficacy (H2b)	0.0036	0.0028	[−0.0002, 0.0106]
Narrative voice → Identification → Cognitive elaboration → Response efficacy (H2c)	**0.0439**	**0.0225**	**[0.0080, 0.0954]**

Note. Narrative voice (independent variable) was dummy coded (0 = third-person narrative, 1 = first-person narrative). We used 95% percentile bootstrap confidence intervals based on 10,000 bootstrap samples for statistical inference of the conditional indirect effects. A specific indirect effect is considered to be statistically significant if the established confidence interval (95% CI) does not include the value 0. If the value 0 is included in the confidence interval, the specific indirect effect is equal to 0, that is, there is no association between the variables involved [45]. Significant specific indirect effects are shown in bold.

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
