# Peer review of "Narrative Voice Matters! Improving Smoking Prevention with Testimonial Messages through Identification and Cognitive Processes"

_ijerph, 2020, doi:10.3390/ijerph17197281_

Round 1

Reviewer 1 Report

This is an interesting study in which the authors tested a narrative message to persuade smokers into quitting. The experimental design consisted of a 2 (narrative voice) × 2 (message) factorial experiment in which 525 adult smokers (ages between 18 and 55) were randomly assigned to a first-person versus third-person narrative message, and two different messages were used within each experimental condition. Two hypotheses were tested, the first (H1) hypothesized that a first-person narrative message will induce greater identification with the protagonist than a third-person message (this hypothesis was confirmed), and the second (H2) hypothesized that the indirect effect of narrative voice on the effectiveness of the message, the intention to quit smoking, the self-efficacy expectations, and the response efficacy expectations is mediated by the identification with the protagonist and by cognitive processes such as counterarguing (H2a, partially supported by the results), reactance (H2b, not supported by the results and therefore rejected), and cognitive elaboration (H2c, supported by the results).

The article is well written, well explained, the results are presented in a logic sequence, is easy to follow, however the whole analysis required several statistical tests that should be clearly explained in the Materials and Methods section, in a separated subsection.

Probably the authors want to discuss further how this approach (using an appropriate narrative message in first-person) could be implemented into smoking prevention programmes (or even prevention of other addictions), and potential limitations or concerns regarding the different cognitive processes that may increase or reduce the success of the programme.

Section 2.2: Independent variable and stimulus materials, indicate somewhere that this information is shown as Supplementary material.

Author Response

Response to Reviewer 1

We thank the Reviewer for his/her close attention to the manuscript and for all the suggestions. Thank you very much for your very positive comments regarding our manuscript: “This is an interesting study in which the authors tested a narrative message to persuade smokers into quitting” and “The article is well written, well explained, the results are presented in a logic sequence, is easy to follow.”

Thank you for bringing important theoretical and methodological issues to our attention. We carefully attended to each of the issues, as detailed below.

Comment # 1.- The article is well written, well explained, the results are presented in a logic sequence, is easy to follow, however the whole analysis required several statistical tests that should be clearly explained in the Materials and Methods section, in a separated subsection.

Response # 1.- Thank you for your insightful feedback. We have added information on the statistical approaches in the revision, in a separate subsection (2.4. Statistical Analysis).

Data analyses were conducted using IBM SPSS 25 statistical software. Descriptive analysis (means and standard deviations) were calculated to examine sample demographics (see Table 1). Reliability (Cronbach’s alpha) was calculated for all the measures (see Table 2). The correlations between the mediating and dependent variables were analysed by using the Pearson correlation coefficient. One-way analysis of variance (ANOVA) and chi-squared test were used to test the success of randomization. Factorial ANOVA was performed to determine the impact of narrative voice on identification with the protagonist (H1), including the type of message as a second independent variable. Effect size in ANOVA test was calculated using partial eta-squared (partial h2); for nonsignificant results (p-values higher than 0.05), effect size was substituted by observed power (or post hoc power), as recommended by many statisticians (e.g., Onwugbuzie & Leech, 2004; but, also see, O’Keefe, 2007). To test hypothesis 2, the PROCESS macro (version 3.5) for SPSS developed by Hayes (2018) was used. This macro makes it possible to test different mediational models based on the bootstrapping technique. According to the bootstrapping method, an indirect effect is statistically significant if the confidence interval established (CI at 95%) does not include the value 0. If the value 0 is included in the CI, the indirect effect is equal to 0, that is, there is no association between the variables considered.

[Table 1, Study participant characteristics, was requested by reviewer 2].

References:

Hayes, A. F. (2018). Introduction to mediation, moderation, and conditional process analysis. New York, NY: The Guilford Press (2nd edition).

O’Keefe, D. J. (2007). Brief report: post hoc power, observed power, a priori power, retrospective power, prospective power, achieved power: sorting out appropriate uses of statistical power analyses. Communication Methods and Measures, 1(4), 291-299. doi: 10.1080/19312450701641375

Onwuegbuzie, A. J., & Leech, N. L. (2004). Post hoc power: a concept whose time has come. Understanding Statistics, 3(4), 201-230. doi: 10.1207/s15328031us0304_1

Comment # 2.- Probably the authors want to discuss further how this approach (using an appropriate narrative message in first-person) could be implemented into smoking prevention programmes (or even prevention of other addictions), and potential limitations or concerns regarding the different cognitive processes that may increase or reduce the success of the programme.

Response # 2.- Thank you for this observation. We have added the following (final) comment in the Conclusions of the manuscript:

The results obtained in our research suggest various applications in the field of prevention and treatment of smoking. We consider that a narrative intervention such as that proposed (through narrative messages written in the first person, with testimonies of former smokers who relate their successful experience) could constitute a primary prevention tool, since any attempt to reduce tobacco consumption indirectly seeks to avoid the development of associated health problems. This type of testimonial messages (relating the experience of someone who has overcome tobacco addiction) would thus serve both to prevent smoking (stopping people from starting to use tobacco) and to help active smokers to quit tobacco, thus avoiding the damage that it may cause to them. In this sense, testimonial messages of former smokers could be used in prevention campaigns and be disseminated through social networks (for example, Instagram or Facebook), or as support materials in broader campaigns disseminated on web pages.

Comment # 3.- Probably the authors want to discuss further how this approach (using an appropriate narrative message in first-person) could be implemented into smoking prevention programmes (or even prevention of other addictions), and potential limitations or concerns regarding the different cognitive processes that may increase or reduce the success of the programme.

Response # 3.- Thank you for this useful comment. We have added the following comment to the Discussion of the manuscript:

The most important limitation of this work is that the proposed mediators were measured rather than being experimentally manipulated, which prevents conclusions with total certainty regarding the proposed causal sequence (identification --> counterarguing --> outcomes; identification --> reactance --> outcomes; identification --> cognitive elaboration --> outcomes). This problem is present in studies that contrast serial mediational models in this field (e.g., Dunlop, Wakefield & Kashima, 2010; Kim & Lee, 2017). Although temporal precedence is an important element to establish a causal inference, it is also necessary to propose a theoretical argument for the relationship between the mediating mechanisms, a condition that our work fulfils by relying on theoretical models of narrative persuasion. Indeed, future research should use other methodological approaches to deal with such causal inference problems (Pirlott & MacKinnon, 2016).

References:

Dunlop, S. M., Wakefield, M., & Kashima, Y. (2010). Pathways to persuasion: cognitive and experiential responses to health-promoting mass media messages. Communication Research, 37(1), 133–164. doi: 10.1177/0093650209351912

Kim, H. K., & Lee, T. K. (2017). Conditional effects of gain–loss-framed narratives among current smokers at different stages of change. Journal of Health Communication, 22(12), 990-998. doi: 10.1080/10810730.2017.1396629

Pirlott, A. G., & MacKinnon, D. P. (2016). Design approaches to experimental mediation. Journal of Experimental Social Psychology, 66, 29–38. doi: 10.1016/j.jesp.2015.09.012

Comment # 4.- Section 2.2: Independent variable and stimulus materials, indicate somewhere that this information is shown as Supplementary material.

Response # 4.- Thank you for your insightful feedback. We have added information (text in blue) regarding the supplementary material in the first paragraph of Section 2.2. Independent variable and stimulus materials:

Taking as examples the testimonies of former smokers on forums and web pages as well as narratives used in previous studies [28], a written narrative delivered by a 45-year-old woman who indicated that she had quit smoking more than a year ago was constructed (narrative messages are available in the supplementary material file online). In her story, the protagonist alluded to topics such as the age at which she started smoking, why she considered quitting smoking (after having being diagnosed with periodontitis) and how she managed to quit tobacco, commenting in the final part of the narrative that she no longer wanted to start smoking again, that she had noted how the negative consequences associated with tobacco use had disappeared and that she was experiencing a number of benefits from quitting smoking.

Reviewer 2 Report

This study investigated the impact of narrative voice (first person, third person) in persuasive messages about smoking cessation in an online sample of adult cigarette smokers. The idea and study concept are interesting and it potentially adds value to the research on tobacco control communication messaging. I have several comments/questions for the authors’ consideration, listed below by section of the manuscript.

Abstract

It would be helpful to specify the direction of some of the observed findings in the abstract. As written, it only describes associations but not which direction the associations run.

Introduction

The introduction is appropriately concise, but I think it is missing 3 important elements:

1) Why is the study focusing on tobacco/smoking? There is no public health context, and while this may be obvious to many a few sentences to characterize the importance of tobacco as a public health problem and the role of communication campaigns as an intervention strategy.

2) What is the evidence on narrative communications in tobacco control, and what are the limitations?

3) Why did you select outcome measures of self-efficacy, response efficacy, perceived effectiveness (this isn’t described later, until the methods), and intentions to quit? Are these derived from the theoretical framework described? Or chosen for some other reason(s), theoretical (e.g., EPPM for efficacy beliefs) or empirical? The introduction needs to include some rationale to support the outcome variables chosen.

Methods

The methods are generally well-written. I have a few specific comments/questions:

1) Page 3, I suggest avoiding using the word “habitual” to describe smokers. For eligibility criteria of 100 cigarettes lifetime and now smoking 5 or more cigarettes daily within the past week, how were these determined?

2) Page 3, second paragraph, the paper describes 43 participants removed due to failing the manipulation test. In the 4th paragraph, the paper discusses quality controls including a “control” question. Are these the same things repeated, or 2 different sets of quality control where participants were removed?

3) The two messages tested in the 1st and 3rd person narrative conditions differ by only a few words, specifying in one message three prior quit attempts and in the other a first attempt to quit. It is unclear how this modification, which is extremely minor, increases external validity since in typical tobacco control communication messaging campaigns the content varies much more widely from message to message.

Results

1) It would be helpful to include a table describing the characteristics of the sample (sociodemographics, smoking history, use of other tobacco products, pre-exposure measures).

2) I’m not sure how to interpret the “observed power” statistics provided on page 7 in the results for ANOVA findings. Some description/clarification is needed.

Discussion

The discussion is generally well-written and concise. A few comments/questions

1) Third paragraph, I am not sure how well the text describing the serial mediation model aligns with the data. For example, this paragraph seems to describe significant indirect paths from narrative voice, through reduced counter arguing, that greater identification is associated with all outcomes examined. Only 2 of these indirect paths are significant in Table 3 (Narrative to identifcation to counterarguing to response efficacy; narrative to identification to counterarguing to perceived message effectiveness).

2) The subsequent section indicates “our study reinforces the idea that the experience of fusion with a protagonist of the message becomes a process that is incompatible with a state of negative assessment.” To me this seems to overgeneralize the findings a bit, since reactance could also be considered a negative assessment (or response) but none of the serial mediation processes involving reactance were significant. Can the authors comment on this, or update the statement in the discussion to align better with the data?

Author Response

Response to Reviewer 2

We thank the Reviewer for his/her positive comments about our manuscript, regarding the introduction (“The idea and study concept are interesting and it potentially adds value to the research on tobacco control communication messaging”), method (“The methods are generally well-written”) and discussion (“The discussion is generally well-written and concise”). Finally, we thank you for your detailed suggestions for clarifying and better organizing the manuscript.

Comment # 1.- [Abstract]. It would be helpful to specify the direction of some of the observed findings in the abstract. As written, it only describes associations but not which direction the associations run.

Response # 1.- Thank you for this comment. However, we consider that the final sentence of the Abstract already refers to the direction of the results. In any case, we have slightly modified the wording of the last sentence (in blue) to present more clearly a synthesis of the results obtained:

Abstract: Narrative messages are increasingly being used in the field of tobacco prevention. Our study is based on narrative persuasion and aims to analyse the psychological mechanisms that explain why the narrative voice is relevant to promote persuasive impact. An online experiment with a 2 (narrative voice) × 2 (message) factorial design was carried out. Participants (525 adult smokers) were randomly assigned to two experimental conditions (first-person versus third-person narrative message). To increase the external validity of the study, two different messages were used within each condition. After reading the narrative message the mediating and dependent variables were evaluated. Participants who read the narrative in the first person experienced greater identification. Moreover, mediational analysis showed that both counterarguing and cognitive elaboration played a significant role in the relationship between narrative voice, identification and persuasive impact. This study confirms that narrative voice is not only an anecdotal formal choice, but that it indirectly affects variables related to tobacco prevention, due to the fact that first-person messages activate a mechanism of affective connection with the message (increasing the identification with the protagonist) that decreases resistance to prevention (the counterarguing process) while simultaneously stimulating reflection or cognitive elaboration.

Comment # 2.- [Introduction]. Why is the study focusing on tobacco / smoking? There is no public health context, and while this may be obvious to many a few sentences to characterize the importance of tobacco as a public health problem and the role of communication campaigns as an intervention strategy.

Response # 2.- Thank you for this useful comment. We have added the following comment to the Introduction of the manuscript:

Tobacco is one of the main public health problems facing humanity. The consumption of this substance has been linked to multiple health problems (such as respiratory and cardiovascular diseases and various forms of cancer), being responsible for more than 8 million deaths a year (American Cancer Society, 2018; World Health Organization, 2020). In Spain, 18.8% of women and 25.6% of men smoke daily, constituting the second most widespread psychoactive substance in the population and being responsible for 50,000 deaths per year (13% of all deaths) (AECC, 2018). However, smoking is also the largest preventable cause of premature death and illness (US Department of Health and Human Services, 2014). Therefore, improving the effectiveness of smoking prevention campaigns or campaigns aimed at helping smokers to quit is an important health communication goal.

The design of campaigns for the prevention of smoking requires innovative approaches that consider both the characteristics of the messages and the psychological processes they trigger. In this sense, narrative messages are increasingly being used in the field of tobacco prevention [1, 2] ...

References:

American Cancer Society (2018). Health risks of smoking tobacco. Retrieved from https://www.cancer.org/cancer/cancer-causes/tobacco-and-cancer/health-risks-of-smoking-tobacco.html#references

AECC (2018) Tabaquismo y cáncer en España. Situación actual. Available online:  https://www.aecc.es/sites/default/files/content-file/Informe-tabaquisimo-cancer-20182.pdf (accessed on 18 September 2020)

World Health Organization (2020). Tobacco. Retrieved from http://www.who.int/mediacentre/factsheets/fs339/en/

U.S. Department of Health and Human Services (2014). The health consequences of smoking: 50 years of progress. A report of the Surgeon General. Atlanta, GA: U.S. Department of Health and Human Services, Centers for Disease Control and Prevention, National Center for Chronic Disease Prevention and Health Promotion, Office on Smoking and Health. Retrieved from https://www.hhs.gov/sites/default/files/consequences-smoking-exec-summary.pdf

Comment # 3.- [Introduction]. What is the evidence on narrative communications in tobacco control, and what are the limitations?

Response # 3.- Thank you for this observation. A series of modifications have been made to the Introduction of the manuscript:

First, the following text (blue text) has been added as a fourth paragraph in the initial version of the manuscript in the Introduction:

Despite being a very relevant formal resource for the design of this type of message (Chen, Bell, & Taylor, 2016; de Graaf et al., 2016), analysis of the effect of narrative voice has received insufficient attention in research on narrative persuasion for smoking control. For example, in a review of 153 experimental studies on health-related narrative persuasion, only 4 manipulated this attribute of the message, and only 1 of them focused on smoking prevention (de Graaf et al., 2016). Our work therefore aims to contribute to knowledge on the impact of narrative voice in smoking prevention, as this line of research has been developed very little to date.

In addition, the wording of the fifth paragraph in the initial version of the manuscript (text in blue) has been modified as follows:

Previous research has found that people better imagine the thought processes of protagonists (e.g., “Did you see what the main character was thinking and seeing?”) when narrative messages are written in the first compared with the third person (Segal, Miller, Hosenfeld, Mendelsohn, Russell, Julian, Greene, & Delphonse, 1997). Therefore, it has been proposed that first-person narratives make it easier for the audience to identify with the protagonist [9, 16, 17]. In this context, identification with the protagonist (or the capacity to feel and adopt the point of view of the protagonist of the narrative; [7, 18, 19] is proposed as a primary mediator.

Finally, a slight modification has been made (text in blue) in what was the eighth paragraph of the initial version of the manuscript:

In this context, it is assumed that identification can facilitate persuasive impact through three cognitive processes that would act as secondary mediating mechanisms. Consistent with the Entertainment Overcoming Resistance Model (EORM) [14], identification can be expected to reduce counterarguing (the production of critical cognitive responses that refute the content of the message; [22]) and reactance (negative reactions to the message when perceiving that freedom of choice or opinion is being threatened; [23]). Furthermore, it may be expected that identification will be associated with an increase in cognitive elaboration (reflecting on the topic of the message during its processing) [21, 24]. However, to date, the role of counterarguing, reactance and cognitive elaboration has not been analysed simultaneously within a single mediational model, so our study constitutes an original contribution to this field. Therefore, our work tries to contrast the role of these three cognitive mechanisms to explain the indirect effect of narrative voice on measures related to the prevention of smoking, leading to the following mediational hypothesis (Figure 1):

Comment # 4.- [Introduction]. Why did you select outcome measures of self-efficacy, response efficacy, perceived effectiveness (this isn't described later, until the methods), and intentions to quit? Are these derived from the theoretical framework described? Or chosen for some other reason(s), theoretical (e.g., EPPM for efficacy beliefs) or empirical? The introduction needs to include some rationale to support the outcome variables chosen.

Response # 4.- Thank you for bringing this important theoretical and methodological issue to our attention. The following text has been added to the first paragraph of the initial version of the manuscript of the Introduction:

The ultimate goal of these types of messages is to cause a persuasive impact. In the present work we use the expression persuasive impact to refer to four outcome measures: intention to quit smoking, perceived effectiveness of the message, expectations of self-efficacy (confidence in one’s ability to abstain from smoking) and expectations of the efficacy of the preventive response (response efficacy or outcome expectations). These measures constitute relevant variables in the theoretical models on health communication and behavioral change (e.g., theory of planned behavior, theory of reasoned action, health belief model, social cognitive theory and extended parallel process model; for a review, see Viswanath, Wallington, & Blake, 2009). The mentioned theoretical models “are frequently applied to the crafting of persuasive health messages and campaigns” (Myrick, 2019, p. 311). Furthermore, these measures have been used as dependent variables in several previous studies on smoking prevention using narrative messages (e.g., Dunlop, Wakefield & Kashima, 2010; Kim, 2019; Kim, Shi, & Cappella, 2016).

References:

Dunlop, S. M., Wakefield, M., & Kashima, Y. (2010). Pathways to persuasion: cognitive and experiential responses to health-promoting mass media messages. Communication Research, 37(1), 133–164. doi: 10.1177/0093650209351912

Kim, M. (2019). When similarity strikes back: conditional persuasive effects of character-audience similarity in anti-smoking campaign. Human Communication Research, 45(1), 52–77. doi: 10.1093/hcr/hqy013

Kim, M., Shi, R., & Cappella, J. N. (2016). Effect of character–audience similarity on the perceived effectiveness of antismoking PSAs via engagement. Health Communication, 31(10), 1193–1204. doi: 10.1080/10410236.2015.1048421

Myrick, J. G. (2019). Media effects and health. In M. B. Oliver, A. A. Raney & J. Bryant (Eds.), Media Effects. Advances in theory and research (pp. 308-323). New York, NY: Routledge.

Viswanath, K., Wallington, S. F., & Blake, K. D. (2009). Media effects and population health. In R. L. Nabi & M. B. Oliver (Eds.), The SAGE handbook of media processes and effects (pp. 313-329). Thousand Oaks, CA: Sage.

Comment # 5.- [Method]. Page 3, I suggest avoiding using the word “habitual” to describe smokers. For eligibility criteria of 100 cigarettes lifetime and now smoking 5 or more cigarettes daily within the past week, how were these determined?

Comment # 5.- Thank you for this observation. The following modifications have been made to the third paragraph of the Design and Participants subsection of the manuscript:

First, the term “habitual” has been changed to “current smokers”.

In addition, the reference values on tobacco consumption used to select the sample of current smokers were taken from previous studies (Kim, 2019; Kim, Bigman, Leader, Lerman, & Cappella, 2012; Kim & Lee, 2017).

The text has thus been modified as follows:

Sociodemographic information was collected in the pre-test measures, and three screener questions were included to select the participants, such that only those who declared that they were current smokers, indicating having smoked more than 100 cigarettes during their life and 5 or more cigarettes each day during the previous week, were included. These eligibility criteria were used with reference to previous studies (Kim, 2019; Kim, Bigman, Leader, Lerman, & Cappella, 2012; Kim & Lee, 2017).

References:

Kim, M. (2019). When similarity strikes back: conditional persuasive effects of character-audience similarity in anti-smoking campaign. Human Communication Research, 45(1), 52–77. doi: 10.1093/hcr/hqy013

Kim, H. S., Bigman, C. A., Leader, A. E., Lerman, C., & Cappella, J. N. (2012). Narrative health communication and behavior change: the influence of exemplars in the news on intention to quit smoking. Journal of Communication, 62(3), 473–492. doi: 10.1111/j.1460-2466.2012.01644.x

Kim, H. K., & Lee, T. K. (2017). Conditional effects of gain–loss-framed narratives among current smokers at different stages of change. Journal of Health Communication, 22(12), 990-998. doi: 10.1080/10810730.2017.1396629

Comment # 6.- [Method]. Page 3, second paragraph, the paper describes 43 participants removed due to failing the manipulation test. In the 4th paragraph, the paper discusses quality controls including a “control” question. Are these the same things repeated, or 2 different sets of quality control where participants were removed?

Response # 6.- Thank you for this observation. Actually, these are two different things. The control for the effectiveness of the experimental manipulation is described in the second paragraph of the Design and Participants section. On the other hand, when designing the questionnaire and fieldwork with QUALTRICS, several measures were implemented to guarantee quality control of the data collection of the participants, which are described in the fourth paragraph of the mentioned section (for example, the inclusion of an attentional control question and several measures related to the time taken to complete the questionnaire or to read the narrative message). We have modified the wording of the final paragraph of the Design and Participants section of the manuscript, as follows:

QUALTRICS allows the implementation of a series of quality controls. The questionnaire was designed in such a way that it could only be completed in a single session. In addition, only those participants who took 6–45 minutes to complete the questionnaire (M = 10.95 minutes, SD = 4.99), took between 60 and 420 seconds to read the narrative (M = 112.88, SD = 53.58) and correctly answered an attention check question included in the questionnaire were counted as valid cases.

Comment # 7.- [Method]. The two messages tested in the 1st and 3rd person narrative conditions differ by only a few words, specifying in one message three prior quit attempts and in the other a first attempt to quit. It is unclear how this modification, which is extremely minor, increases external validity since in typical tobacco control communication messaging campaigns the content varies much more widely from message to message.

Response # 7.- Thank you for this comment. The manipulation of the narrative voice was carried out following standardized procedures in this field. In order to show more clearly how the manipulation of the narrative voice was carried out, the elements that mark the grammatical person (such as the choice of personal pronouns) have been marked (in bold) in the narrative messages (see new version of the supplementary file). This experimental procedure to manipulate the narrative voice has been used successfully in a large number of investigations (e.g., Banerjee & Greene, 2012; Chen, Bell, & Taylor, 2017; Chen, McGlone & Bell, 2015; Nan, Dahlstrom, Richards, & Rangarajan, 2015).

On the other hand, while it is true that research in Media Psychology has emphasized that it is necessary to use more than one message per experimental condition, this practice is still very infrequent. Reeves, Yeykelis and Cummings (2016) reviewed more than 300 experiments conducted in the last 10 years in this field and concluded that “the majority used only a single stimulus per condition (64.4%)” (p. 57). Furthermore, in most experiments carried out in health communication from the perspective of narrative persuasion, only a single narrative message is used per experimental condition (e.g., Kim & Lee, 2017). Indeed, the messages used in campaigns aimed at combating smoking (e.g., Tips from Former Smokers) vary in a quite a large number of their elements. However, we believe that considering all the elements that determine the differences between narrative messages is not feasible in a single investigation, or would require the use of a highly complex experimental design, thus leading to a considerable increase in the sample size and study budget. For example, Kim (2019) needed access to a sample of 1,843 adult smokers when using an experimental design that manipulated the similarity with the protagonist of the message, the theme of the message, and the severity of the consequences derived from smoking. Based on all those considerations, we believe that the inclusion of two different messages per experimental condition increases the external validity of our study with respect to previous research in this field and that it thus represents a relevant methodological contribution.

References:

Banerjee, S. C., & Greene, K. (2012). Role of transportation in the persuasion process: cognitive and affective responses to antidrug narratives. Journal of Health Communication, 17(5), 564–581. Doi: 10.1080/10810730.2011.635779

Chen, M., Bell, R. A., & Taylor L. D. (2017). Persuasive effects of point of view, protagonist competence, and similarity in a health narrative about type 2 diabetes. Journal of Health Communication, 22(8), 702–712. Doi: 10.1080/10810730.2017.1341568

Chen, M., McGlone, M. S., & Bell, R. A. (2015). Persuasive effects of linguistic agency assignments and point of view in narrative health messages about colon cancer. Journal of Health Communication, 20(8), 977–988. Doi: 10.1080/10810730.2015.1018625

Nan, X., Dahlstrom, M. F., Richards, A., & Rangarajan, S. (2015). Influence of evidence type and narrative type on HPV risk perception and intention to obtain the HPV vaccine. Health Communication, 30(3), 301–308. Doi: 10.1080/104102336.2014.888629

Reeves, B., Yeykelis, L., & Cummings, J. J. (2016). The use of media in media psychology. Media Psychology, 19(1), 49-71. Doi: 10.1080/15213269.2015.1030083

Comment # 8.- [Results]. It would be helpful to include a table describing the characteristics of the sample (sociodemographics, smoking history, use of other tobacco products, pre-exposure measures).

Response # 8.- Thank you for this useful comment. We have incorporated a new table (Table 1) with some basic information on the participants in the study. In addition, we have added the following text (see Table 1), at the end of the second paragraph of the Design and Participants subsection:

The online experiment was carried out using QUALTRICS to access an initial sample of 568 adult smokers. Of these, 43 were removed for failing the manipulation test (correctly remembering which type of message they had read: first or third person). The remaining sample consisted of 525 participants (50.9% women). Participant age ranged from 18 to 55 years old (M = 35.27, SD = 10.97) (see Table 1).

Table 1. Characteristics of the study participants (N = 525)

Mean (SD) or percentage

Range

Age

M = 35.27

18–55

SD = 10.97

Sex

Male: 258 (49.1%)

Female: 267 (50.9%)

Fagerström test

M = 4.65

0–10

SD = 2.23

Comment # 9.- [Results]. I’m not sure how to interpret the “observed power” statistics provided on page 7 in the results for ANOVA findings. Some description/clarification is needed.

Response # 9.- Section 2.4 Statistical Analysis (requested by Reviewer 1) has been added, where the statistical procedures used are described, and a brief mention is made of this issue (see text in blue):

Data analyses were conducted using IBM SPSS 25 statistical software. Descriptive analysis (means and standard deviations) were calculated to examine sample demographics (see Table 1). Reliability (Cronbach’s alpha) was calculated for all the measures (see Table 2). The correlations between the mediating and dependent variables were analysed by using the Pearson correlation coefficient. One-way analysis of variance (ANOVA) and chi-squared test were used to test the success of randomization. Factorial ANOVA was performed to determine the impact of narrative voice on identification with the protagonist (H1), including the type of message as a second independent variable. Effect size in ANOVA test was calculated using partial eta-squared (partial h2); for nonsignificant results (p-values higher than 0.05), effect size was substituted by observed power (or post hoc power), as recommended by many statisticians (e.g., Onwugbuzie & Leech, 2004; but, also see, O’Keefe, 2007). To test hypothesis 2, the PROCESS macro (version 3.5) for SPSS developed by Hayes (2018) was used. This macro makes it possible to test different mediational models based on the bootstrapping technique. According to the bootstrapping method, an indirect effect is statistically significant if the confidence interval established (CI at 95%) does not include the value 0. If the value 0 is included in the CI, the indirect effect is equal to 0, that is, there is no association between the variables considered.

References:

Hayes, A. F. (2018). Introduction to mediation, moderation, and conditional process analysis. New York, NY: The Guilford Press (2nd edition).

O’Keefe, D. J. (2007). Brief report: post hoc power, observed power, a priori power, retrospective power, prospective power, achieved power: sorting out appropriate uses of statistical power analyses. Communication Methods and Measures, 1(4), 291-299. doi: 10.1080/19312450701641375

Onwuegbuzie, A. J., & Leech, N. L. (2004). Post hoc power: a concept whose time has come. Understanding Statistics, 3(4), 201-230. doi: 10.1207/s15328031us0304_1

Comment # 10.- [Discussion]. Third paragraph, I am not sure how well the text describing the serial mediation model aligns with the data. For example, this paragraph seems to describe significant indirect paths from narrative voice, through reduced counterarguing, that greater identification is associated with all outcomes examined. Only 2 of these indirect paths are significant in Table 3 (Narrative to identification to counterarguing to response efficacy; narrative to identification to counterarguing to perceived message effectiveness).

Response # 10.- Thank you for detecting this problem. We greatly appreciate the thoughtful analysis of our work. Indeed, only two specific indirect effects were observed, as collated in Table 3. The indicated paragraph has been revised and the wording corrected as follows:

On the one hand, through reduced counterarguing, the participants who identified more with the protagonist showed a more favourable reaction to the message and manifested a greater perceived efficacy of the preventive response (considering that quitting smoking would improve their personal health in the short and long term).

Comment # 11.- [Discussion]. The subsequent section indicates “our study reinforces the idea that the experience of fusion with a protagonist of the message becomes a process that is incompatible with a state of negative assessment.” To me this seems to overgeneralize the findings a bit, since reactance could also be considered a negative assessment (or response) but none of the serial mediation processes involving reactance were significant. Can the authors comment on this, or update the statement in the discussion to align better with the data?

Response # 11.- Thank you for this insightful feedback. We agree with this assessment since reactance did not constitute a significant mediating mechanism. Despite this, the bivariate correlations did show a pattern consistent with our predictions: reactance was negatively associated with identification (r = -.21, p = .000) and with cognitive elaboration (r = -.15, p = .000), and positively with counterarguing (r = .43, p = .000). The following text (marked in blue) has been added to the indicated paragraph:

In this way, our study reinforces the idea that the experience of fusion with the protagonist of the message becomes a process that is incompatible with a state of negative assessment (that would hinder prevention). However, it should be considered that reactance did not act as a significant mediating mechanism, despite its negative correlation with identification (r = -.21, p = .000) and with cognitive elaboration (r = - .15, p = .000), and a positive, strong and statistically significant correlation with counterarguing (r = .43, p = .000).

Round 2

Reviewer 2 Report

I appreciate the authors' prompt and thorough responses. Their revisions addressed all my suggestions/comments.